# Preparation and Electrochemical Properties of LiNi_2/3_Co_1/6_Mn_1/6_O_2_ Cathode Material for Lithium-Ion Batteries

**DOI:** 10.3390/ma14071766

**Published:** 2021-04-02

**Authors:** Meijie Zhu, Jiangang Li, Zhibei Liu, Li Wang, Yuqiong Kang, Zhaohan Dang, Jiasen Yan, Xiangming He

**Affiliations:** 1Beijing Key Laboratory of Fuels Cleaning and Advanced Catalytic Emission Reduction Technology, School of Chemical Engineering, Beijing Institute of Petrochemical Technology, Beijing 102617, China; 2018520012@bipt.edu.cn (M.Z.); 2018520009@bipt.edu.cn (Z.L.); 2019520039@bipt.edu.cn (Z.D.); 2019520009@bipt.edu.cn (J.Y.); 2Institute of Nuclear and New Energy Technology, Tsinghua University, Beijing 100084, China; 3Graduate School at Shenzhen, Tsinghua University, Shenzhen 518055, China; kang.yuqiong@sz.tsinghua.edu.cn; 4Wuxi Vfortune New Energy Battery Materials Co., Ltd., Wuxi, Jiangsu 214135, China

**Keywords:** lithium ion batteries, cathode material, LiNi_2/3_Co_1/6_Mn_1/6_O_2_, preparation, rheological phase method

## Abstract

The cathode material LiNi_2/3_Co_1/6_Mn_1/6_O_2_ with excellent electrochemical performance was prepared successfully by a rheological phase method. The materials obtained were characterized by X-ray diffraction, scanning electron microscopy, electrochemical impedance spectroscopy and charge-discharge tests. The results showed that both calcination temperatures and atmosphere are very important factors affecting the structure and electrochemical performance of LiNi_2/3_Co_1/6_Mn_1/6_O_2_ material. The sample calcinated at 800 °C under O_2_ atmosphere displayed well-crystallized particle morphology, a highly ordered layered structure with low defects, and excellent electrochemical performance. In the voltage range of 2.8–4.3 V, it delivered capacity of 188.9 mAh g^−1^ at 0.2 C and 130.4 mAh g^−1^ at 5 C, respectively. The capacity retention also reached 93.9% after 50 cycles at 0.5 C. All the results suggest that LiNi_2/3_Co_1/6_Mn_1/6_O_2_ is a promising cathode material for lithium-ion batteries.

## 1. Introduction

With the rapid development of portable electronic products and electric vehicles, higher requirements have been placed on the energy density, safety, cycle life, and cost of lithium-ion batteries (LIBs). The nickel-rich ternary layered material LiNi_1-x-y_Co_x_Mn_y_O_2_ such as LiNi_0.8_Co_0.1_Mn_0.1_O_2_(NCM811) and LiNi_0.6_Co_0.2_Mn_0.2_O_2_(NCM622) exhibit high capacity and low cost, showing a promising application prospect [1,2]. However, with the increase of nickel content, the cycle performance, thermal stability, and safety gradually decrease [3,4] due to the factors such as surface residual alkali, transition metal dissolution, cation mixing, surface irreversible formation of NiO phases, intergranular cracks and micro-strains [1,2,3,4,5,6,7,8,9]. Among the nickel-rich LiNi_1-x-y_Co_x_Mn_y_O_2_ materials, NCM622 material can be prepared in the air, and has a higher lithium ion diffusion coefficient and better structural stability [10]; therefore, it has been commercialized and applied on a large scale. For other nickel-rich ternary materials with Ni content higher than 0.6, such as LiNi_0.7_Co_0.15_Mn_0.15_O_2_ and NCM811 materials, the application research is still in progress [1,2,11,12,13].

Based on the first-principles computational studies [14,15], LiNi_2/3_Co_1/6_Mn_1/6_O_2_ and LiNi_0.66_Co_0.17_Mn_0.17_O_2_, with almost the same composition, are also considered very promising cathode materials. Compared with other nickel-rich ternary materials such as LiNi_0.7_Co_0.17_Mn_0.08_O_2_ and LiNi_0.8_Co_0.1_Mn_0.1_O_2_, LiNi_0.66_Co_0.17_Mn_0.17_O_2_ has been confirmed to have higher average voltage, and much higher structural stability with 50% lithium extraction [14]. Its structural stability in the delithiation state is even better than LiCoO_2_ [14]. This means that the material has a good prospect in the application of long-life and high-safety lithium-ion batteries in electric vehicles. Kim [15] verified that LiNi_2/3_Co_1/6_Mn_1/6_O_2_ is expected to be synthesized with an almost perfect crystal structure with few point defects other than some oxygen vacancies (V_O_) and cation-mixing (M_Li_) defects. The two kinds of defect can be suppressed easily by controlling the preparation conditions, which means that LiNi_2/3_Co_1/6_Mn_1/6_O_2_ is also easier to prepare. However, there are relatively few experimental studies on the preparation and electrochemical performance of this material. Saavedra-Arias et al. [14] conducted a sol-gel preparation study of LiNi_0.66_Co_0.17_Mn_0.17_O_2_, and the sample calcined at 800 °C in air delivered a capacity of 167 mAh g^−1^ at 1C rate in the voltage range of 2.5–4.5 V, and a capacity retention of 93.8% after 25 cycles. Such a performance did not meet expectations, so it is still necessary to study the preparation and electrochemical performance of the material further.

It is well known that preparation method has an important influence on the structure and electrochemical performance of materials. The rheological phase method, as a simple and effective synthesis route to ensure the mixing of different reactants at the molecular level, has been applied in the preparation of cathode materials such as LiNi_0.65_Co_0.25_Mn_0.1_O_2_ [16], LiNi_1/2_Mn_1/3_V_1/6_O_2_ [17] and LiNi_1/3_Co_1/3_Mn_1/3_O_2_ [18] in recent years. This method enabled the final products with fewer defects and homogeneous structure, thereby effectively improving the electrochemical performance. In this paper, the rheological phase method was used to prepare the promising cathode material LiNi_2/3_Co_1/6_Mn_1/6_O_2_. The present work was to prepare highly ordered layered LiNi_2/3_Co_1/6_Mn_1/6_O_2_ material with few defects by optimizing the calcination temperature and atmosphere based on a first-principles computational study [15], especially the use of oxygen atmosphere. The material obtained showed excellent electrochemical performance compared with some LiNi_1-x-y_Co_x_Mn_y_O_2_ (0.6 ≤ Ni content < 0.7) materials reported in other papers [14,16,19,20,21]. It can deliver a capacity of 188.9 mAh g^−1^ at 0.2 C and 130.4 mAh g^−1^ at 5 C in the voltage range of 2.8–4.3 V, and retain 93.9% of its initial capacity after 50 cycles at 0.5 C.

## 2. Experimental

### 2.1. Preparation of the Samples

In this work, all the chemical reagents were analytically pure, and purchased from Sinopharm Chemical Reagent Co., Ltd. Stoichiometric LiOH·H_2_O, Ni(Ac)_2_·4H_2_O, Co(Ac)_2_·4H_2_O, Mn(Ac)_2_·4H_2_O and citric acid (CA) with the molar ratio of Li:Ni:Co:Mn:CA = 1.06:2/3:1/6:1/6:2.06 were placed in a ball mill tank, and an appropriate amount of deionized water was added. Then the mixture was milled for 5 h at 300 rpm to form a rheological phase precursor. The precursor was dried at 120 °C for 24 h, ground into a fine powder, then pretreated at 600 °C for 6 h in the muffle furnace (TM-0914P, Beijing Ying’an Meicheng Scientific Instrument Co., Ltd., Beijing, China) to generate an intermediate product. The intermediate product was reground and then calcinated at different temperatures (750 °C, 800 °C, 850 °C, and 900 °C) for 12 h, followed by 700 °C for 12 h to yield the final products. A tube furnace (OTF1200X-II, Shenzhen Kejing Zhida Technology Co., Ltd., Shenzhen, China) was used for calcination in an oxygen atmosphere.

### 2.2. Materials Characterization

The structures of the final products were analyzed by using a D/Max-2600-PC X-ray diffractometer (XRD, Rigaku, Tokyo, Japan, with Cu K_α_ radiation, λ = 1.54056 Å,). The microscopic morphologies and particle size distribution of the four samples calcinated at different temperatures (750 °C, 800 °C, 850 °C, and 900 °C) were analyzed by Quanta-400F field-emission scanning electron microscopy (FEI, Hillsboro, OR, USA) and laser particle size analyzer (Winner2006B, Jinan Winner Particle Instrument Stock Co., Ltd. Jinan, China). Transmission emission microscopy (TEM) and high-resolution transmission electron microscopy (HRTEM) for the sample calcinated at 800 °C were measured by an FEI Tcenai G2 F30 (FEI, Hillsboro, OR, USA). The electrochemical properties of as-prepared materials were tested by galvanostatic charge-discharge test using CR2032 coin cells, in which the cathode electrodes comprising 80% active material, 10% Super P, and 10% poly(vinylidene fluoride) (PVdF) were pasted on Al foil, Li-metal chip were used as anode, LiPF_6_ (1 mol L^−1^) dissolved into ethylene carbonate (EC), diethyl carbonate (DEC) and dimethyl carbonate (DMC) with a volume ratio of 1:1:1 was used as the electrolyte, and Celgard 2400 membrane was used as the separator. The coin cells were assembled in a glove box (Etelux Lab2000, Etelux Inert Gas System (Beijing) Co., Ltd., Beijing, China) filled with argon, and were charge-discharged galvanostatically under the cut-off potential of 2.8 V and 4.3 V (vs. Li/Li^+^) at different current densities (1 C = 200 mA g^−1^) using the Land battery system (LANHE CT2001A, Wuhan Jinnuo Electronics Co., Ltd., Wuhan, China). An electrochemical workstation (IM6eX, Zahner Elektrik GmbH & Co. KG, Kronach, Germany) was used to test the electrochemical impedance spectroscopy (EIS) of the four electrode materials calcinated at different temperatures (750 °C, 800 °C, 850 °C, and 900 °C) after galvanostatic charge-discharge tests at 0.2 C for 5 cycles with alternating current (AC) amplitude of 5 mV over the frequency range of 10 mHz~100 KHz.

## 3. Results and Discussion

It is well known that the stoichiometric LiNiO_2_ is difficult to obtain, because a decomposition of LiNiO_2_ to Li_1-x_Ni_1+x_O_2_ occurs during the high-temperature treatment of LiNiO_2_ [22]. This departure from the ideal composition results in partial reduction of Ni ion’s valence from 3 to 2, which causes the “cation mixing” of Li^+^ and Ni^2+^ due to Ni^2+^ ions with an ionic radius (0.69 Å) similar to that of Li^+^ (0.76 Å), thus leading to poor electrochemical performance [1,22]. The use of an oxygen atmosphere is beneficial to suppress the decomposition reaction of LiNiO_2_ at high temperature [23], therefore a well cation-ordered layered LiNiO_2_ and its derivative materials such as LiNi_0.8_Co_0.15_Al_0.05_O_2_ are usually prepared under an oxygen atmosphere [23,24]. For nickel-rich ternary LiNi_1-x-y_Co_x_Mn_y_O_2_ materials, the Co and Mn ions are in the form of Co^3+^ and Mn^4+^ state respectively, while the oxidation state of Ni ions increases with decreasing Mn content [7]. When the Mn content is low, e.g., NCM811, most nickel ions are in the form of Ni^3+^, and therefore oxygen atmospheres are still required during preparation [25]. When the Mn content is high, e.g., LiNi_1/3_Mn_1/3_Co_1/3_O_2_ and LiNi_0.5_Co_0.2_Mn_0.3_O_2_, it can usually be calcined in air to obtain a ternary material with a well-ordered layered structure [26,27]. Even for the NCM622 material, there are many reports that the sample synthesized in air also exhibits well-ordered layered structure and excellent electrochemical performance [28,29,30]. Compared with NCM622, the Ni content in LiNi_2/3_Co_1/6_Mn_1/6_O_2_ is slightly higher, and the kind of atmosphere that can be used needs to be experimentally determined first.

Figure 1 shows the XRD patterns and initial charge-discharge curves of LiNi_2/3_Co_1/6_Mn_1/6_O_2_ materials calcined at 800 °C under different atmospheres. The XRD data obtained can be analyzed using the MDI-JADE 6.5 software package. From Figure 1a, it can be seen that all peaks can be indexed to the layer α-NaFeO_2_ structure with space group R3-m, and no impurity phase appears. Compared with the sample prepared in air, the material prepared under oxygen atmosphere show clearer split double peaks of (006)/(102) and (008/110), and higher ratio of I_(003)_/I_(104)_ (increased from 1.06 to 1.75), indicating less cation mixing and a better layered structure [31,32]. Based on the first-principles computational study, Kim [15] confirmed that some oxygen vacancies (V_O_) and cation-mixing (M_Li_) defects may appear in LiNi_2/3_Co_1/6_Mn_1/6_O_2_. Obviously, the use of oxygen atmosphere can effectively suppress the generation of such defects, thereby improving the cation ordering in the layered material. This will be of benefit to enhance electrochemical performance. As shown in Figure 1b, the material prepared under oxygen atmosphere exhibits smaller electrochemical polarization and higher charge-discharge capacity and coulombic efficiency. Compared with the sample produced in air, its discharge capacity at 0.2 C rate in the range of 2.8–4.3 V is increased from 170.8 mAh g^−1^ to 188.7 mAh g^−1^, and the coulombic efficiency is also increased from 82.1% to 88.3%.

In addition to the atmosphere, the calcination temperature is also an important factor affecting the defects in the nickel-rich ternary LiNi_1-x-y_Co_x_Mn_y_O_2_ materials. The cationic ordering which leads to the 2D structure requires a temperature of 700 °C or more [33], but too high a temperature treatment can result in more defects due to the decomposition of LiNiO_2_ to Li_1-x_Ni_1+x_O_2_ [22]. Figure 2 shows the XRD patterns of LiNi_2/3_Co_1/6_Mn_1/6_O_2_ materials prepared at different calcination temperatures under an oxygen atmosphere. All samples show a single-phase α-NaFeO_2_-type structure. The lattice parameters refined using the MDI-JADE 6.5 software package are listed in Table 1. It is well known that in addition to the intensity ratio of I_(003)_/I_(104)_, the *c/a* value is also used generally to indicate the cation mixing, and a higher ordered layered structure is obtained when *c/a* >4.899 [34]. The material calcinated at 800 °C has the highest values of *c*/*a* (4.960, close to 5) and I_(003)_/I_(104)_ (1.752, much more than 1.2). That means that the LiNi_2/3_Co_1/6_Mn_1/6_O_2_ sample calcinated at 800 °C has the least defects, and highest ordered layered structure.

Figure 3 shows the particle morphology and particle size distribution of LiNi_2/3_Co_1/6_Mn_1/6_O_2_ materials prepared at different calcination temperatures. From the scanning electron microscopy (SEM) images, it can be seen that the sample calcinated at 800 °C also has a highly crystallized particle morphology with smooth surface, small particle size and uniform particle size distribution. Its primary particles size is about 300~500 nm in diameter, but the D_10_, D_50_ and D_90_ obtained from the particle size distribution analysis is 2.1, 2.8, and 3.2 µm, respectively. This means that the primary particles can agglomerate to form secondary particles with a medium particle size of 2.8 µm. Although the secondary particle size is the smallest among the four samples, its primary particles are highly crystalline. This can be further improved by transmission electron microscopy (TEM) and high-resolution TEM (HRTEM) analysis. As shown in Figure 4, the sample calcinated at 800 °C possesses a highly ordered layered structure, and the layer spacing d_(003)_ is 0.47 nm, which is completely consistent with the XRD results. The highly crystalline and well-dispersed particles with small particle size can increase the contact area with the electrolyte, and shorten the path for Li^+^ diffusion inside the particles, thereby helping to provide higher capacity at high rate.

Furthermore, the electrochemical performances of LiNi_2/3_Co_1/6_Mn_1/6_O_2_ materials prepared at different calcination temperatures were investigated. Figure 5 shows the first charge-discharge curves (Figure 5a) at 0.2 C rate and the AC impedance spectroscopies (Figure 5b, Nyquist plots). The Nyquist plots were fitted by using the equivalent circuit (Figure 5b inset), which included electrolyte resistance (*R_e_)*, surface film resistance (*R_f_*), charge transfer (*R_ct_*) resistance, two constant phase element (*CPE1*, *CPE2*), and diffusional components like Warburg impedance (*Wo*) [35]. The lithium ion diffusion coefficient was also calculated from the following formula:(1)$$D=\;\frac{R^{2}T^{2}}{2A^{2}n^{4}F^{4}C^{2}\sigma^{2}}\;$$
where *R* is the gas constant, *T* is the room temperature in the experiment, *A* is the surface area of the electrode, *n* is the number of the electrons per molecule attending the electronic transfer reaction, *F* is the Faraday constant, *C* is the concentration of lithium ion in LiNi_2/3_Co_1/6_Mn_1/6_O_2_ electrode, *σ* is the slope of the line Z’~ω^−1/2^ (shown in Figure 5c), respectively [36]. The fitting results and calculated lithium ion diffusion coefficient *D* values are shown in Table 2. The sample prepared at 800°C delivers a high discharge capacity of 188.7 mAh g^−1^, which consists with its perfect layered structure, small interface impedance (*R_f_* + *R_ct_*), and high lithium ion diffusion coefficient. In contrast, the interfacial impedance of the material prepared at 750 °C increases to 41.29 Ω, while the lithium ion diffusion coefficient and discharge capacity decreases to 1.41×10^−11^ cm^2^ s^−1^ and 176.9 mAh g^−1^ respectively, which may be attributed to the low calcination temperature leading to a relatively imperfect layered structure [33]. For the samples calcinated at 850 °C and 900 °C, the discharge capacities reduce rapidly to 170.9 mAh g^−1^ and 167.7 mAh g^−1^, respectively, accompanied by decreased lithium ion diffusion coefficient and large increased interface impedance, which is mainly due to the increased defects caused by loss of oxygen and lithium at high temperatures [22].

Figure 6 shows the rate capability of LiNi_2/3_Co_1/6_Mn_1/6_O_2_ prepared at different temperatures. It is clear that the sample calcinated at 800 °C still displays higher capacity than other three samples at different rate of 0.2 C to 5 C. It delivers capacity of 188.9, 179.1, 161.7, 148.2, 130.4, and 185.1 mAh g^−1^ at rate of 0.2 C, 0.5 C, 1 C, 2 C, and 5 C, respectively. Moreover, it is impressive that the capacity at 0.2 C over 26 to 30 cycles followed after 5 C charge-discharge cycles still remains 97.9% of its initial discharge capacity at 0.2 C in the first cycle. Furthermore, the cycling performance of sample calcinated at 800 °C was tested at rate of 0.5 C. As shown in Figure 7, the capacity retention reaches 93.9% after 50 cycles, and the coulombic efficiency is close to 100% except for the first cycle, indicating good electrochemical reaction reversibility. The above results reveal that calcination at 800 °C under an oxygen atmosphere is optimized conditions for preparing LiNi_2/3_Co_1/6_Mn_1/6_O_2_ material with excellent rate performance and structural stability during cycling at different rates. 

Table 3 lists the electrochemical performance of some LiNi_1-x-y_Co_x_Mn_y_O_2_ (0.6 ≤ Ni content < 0.7) materials reported in other papers. Compared with LiNi_0.66_Co_0.17_Mn_0.17_O_2_ [14] and LiNi_0.65_Co_0.25_Mn_0.1_O_2_ [16] materials obtained using a similar wet chemical synthesis route, the LiNi_2/3_Co_1/6_Mn_1/6_O_2_ sample calcinated at 800 °C under O_2_ atmosphere in this work exhibits excellent electrochemical performance. This can be mainly attributed to the optimized preparation conditions such as calcination temperature and atmosphere, thereby effectively suppressing appearance of oxygen vacancies (V_O_) and cation-mixing (M_Li_) defects. Compared with the electrochemical performance of the pristine NCM622 material at a cut-off voltage of 4.5 V [19,20,21], the as-prepared LiNi_2/3_Co_1/6_Mn_1/6_O_2_ in this work can not only obtain almost the same capacity at a cut-off voltage of 4.3 V, but also exhibit better cycling stability due to the reduced electrolyte decomposition at a low cut-off voltage (4.3 V). Although the cycling performance of NCM622 materials at high cut-off voltage can be improved by modification such as surface coating and element doping [11,12,19,20,21], this undoubtedly increases the complexity and cost of material manufacturing. 

## 4. Conclusions

The nickel-rich ternary-layered LiNi_2/3_Co_1/6_Mn_1/6_O_2_ material can be successfully prepared by a rheological phase method. Both calcination temperature and atmosphere are very important factors to ensure high cation-ordering and low defects of the product. XRD and SEM analysis results indicated that the sample produced under the optimized calcination conditions of 800 °C and oxygen atmosphere displayed well-crystallized particle morphology, highly ordered layered structure with low defects. Electrochemical performance characterization showed that the sample prepared under the optimized conditions exhibited small interface impedance, high lithium ion diffusion coefficient, and excellent electrochemical performance. In the voltage range of 2.8–4.3 V, it delivered capacity of 188.9 mAh g^−1^ at 0.2 C and 130.4 mAh g^−1^ at 5 C, respectively. The capacity retention also reached 93.9% after 50 cycles at 0.5 C. The results prove that LiNi_2/3_Co_1/6_Mn_1/6_O_2_ is a very promising cathode material for lithium-ion batteries.

## Figures and Tables

**Figure 1 materials-14-01766-f001:**
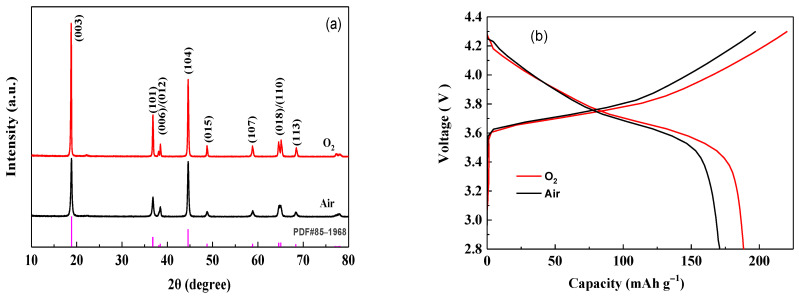
X-ray diffraction (XRD) patterns (**a**) and initial charge-discharge curves at 0.2 C rate (**b**) of LiNi_2/3_Co_1/6_Mn_1/6_O_2_ prepared under different atmosphere.

**Figure 2 materials-14-01766-f002:**
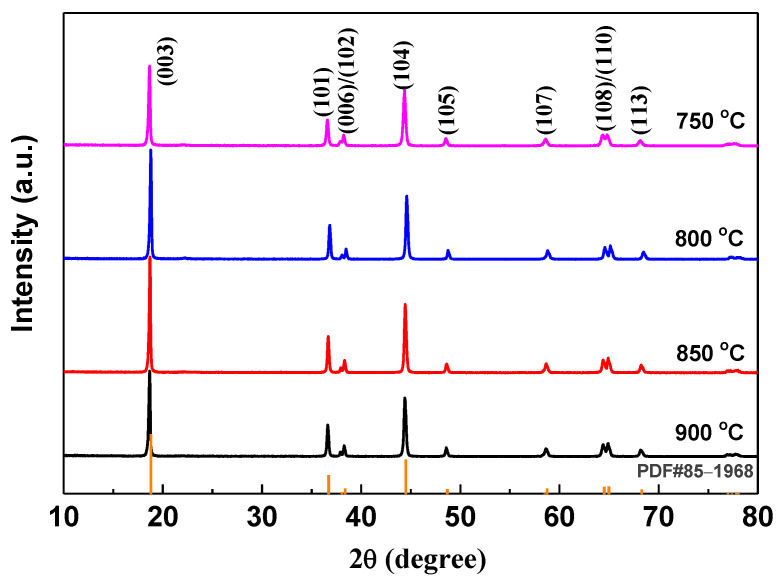
XRD patterns of LiNi_2/3_Co_1/6_Mn_1/6_O_2_ prepared at different temperatures under O_2_ atmosphere.

**Figure 3 materials-14-01766-f003:**
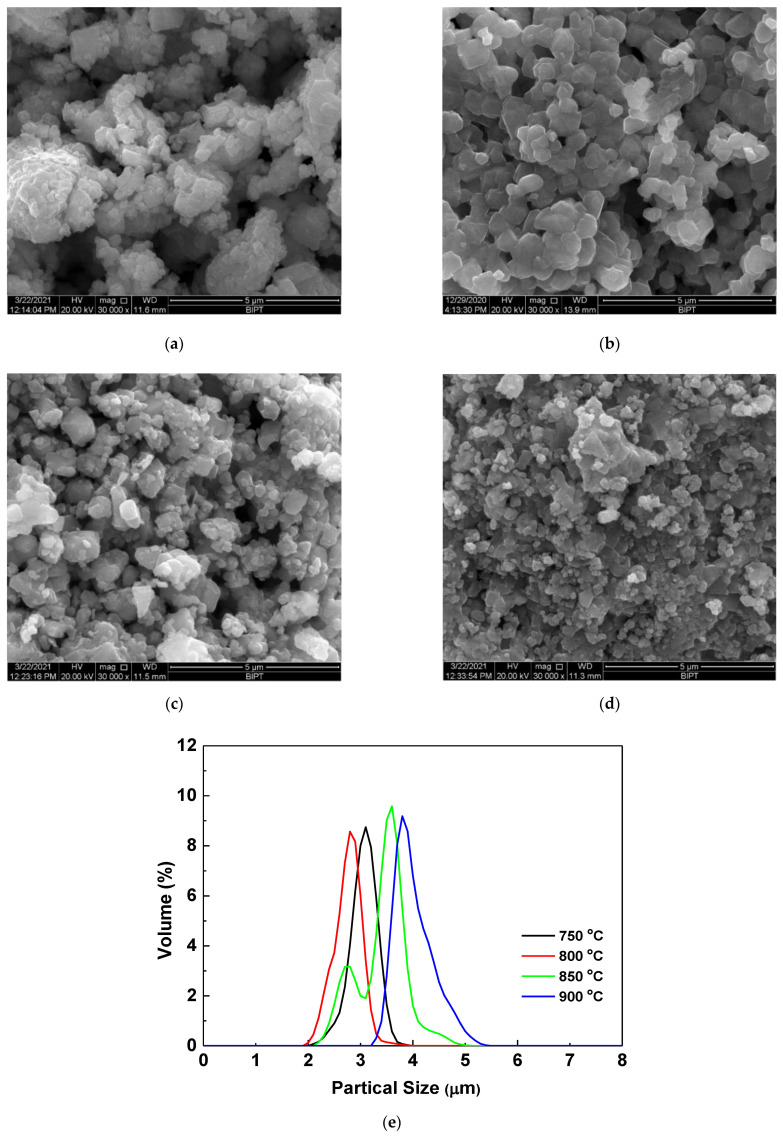
Particle morphology and particle size distribution analysis. (1) Scanning electron microscope (SEM) images of LiNi_2/3_Co_1/6_Mn_1/6_O_2_ prepared at 700 °C (**a**), 800 °C (**b**), 850 °C (**c**) and 900 °C (**d**). (2) Particle size distribution (**e**).

**Figure 4 materials-14-01766-f004:**
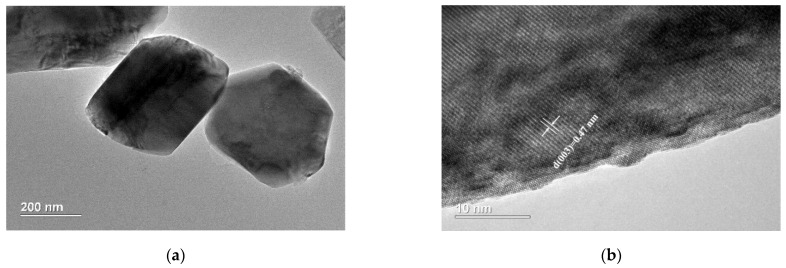
Transmission electron microscope (TEM) image (**a**) and high-resolution TEM (HRTEM) image (**b**) of LiNi_2/3_Co_1/6_Mn_1/6_O_2_ calcinated at 800 °C.

**Figure 5 materials-14-01766-f005:**
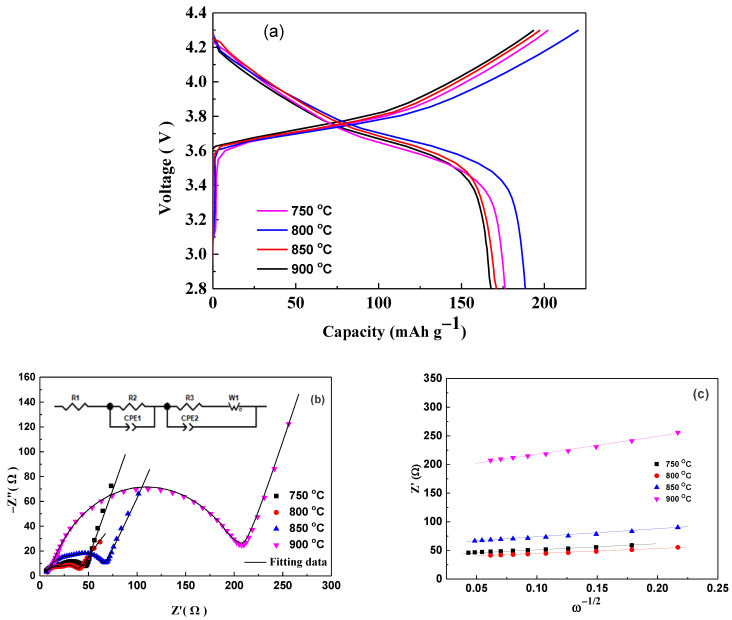
The initial charge-discharge curves (**a**), Nyquist plots (**b**), and Z’ vs. ω^−1/2^ plots (**c**) of LiNi_2/3_Co_1/6_Mn_1/6_O_2_ prepared at different temperatures.

**Figure 6 materials-14-01766-f006:**
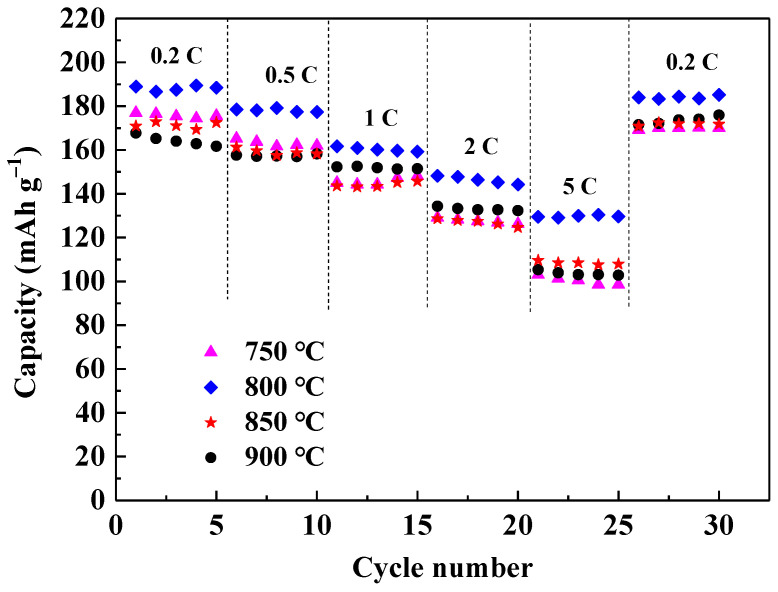
Rate capability of LiNi_2/3_Co_1/6_Mn_1/6_O_2_ calcinated at different temperatures.

**Figure 7 materials-14-01766-f007:**
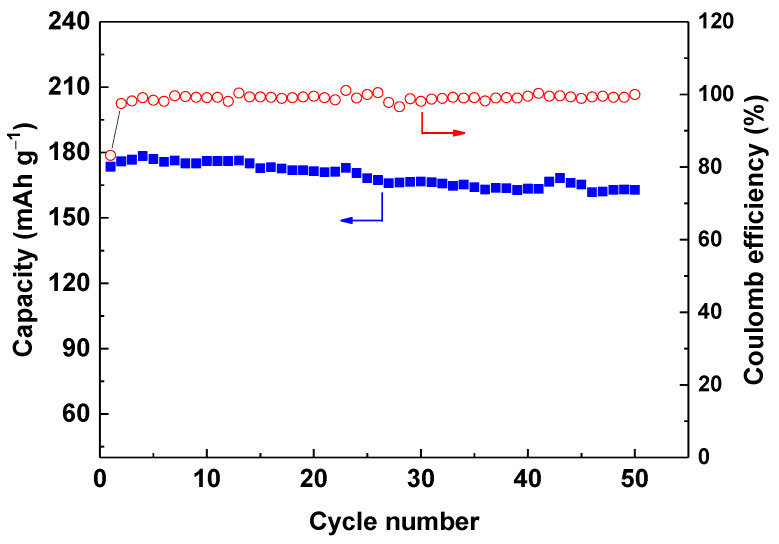
Cycling performance of LiNi_2/3_Co_1/6_Mn_1/6_O_2_ calcinated at 800 °C.

**Table 1 materials-14-01766-t001:** Lattice parameters of LiNi_2/3_Co_1/6_Mn_1/6_O_2_ materials prepared at different calcination temperatures.

Temperature (°C)	*c* (Å)	*a* (Å)	*c*/*a*	*V* (Å^3^)	I_(003)_/I_(104)_
750	14.2134	2.8761	4.9419	101.82	1.43
800	14.1848	2.8601	4.9596	100.48	1.75
850	14.2134	2.8704	4.9517	101.41	1.70
900	14.2134	2.8730	4.9471	101.60	1.46

**Table 2 materials-14-01766-t002:** Electrochemical impedance spectroscopy (EIS) analysis results of LiNi_2/3_Co_1/6_Mn_1/6_O_2_ prepared at different temperatures.

T (°C)	*R_s_* (Ω)	*R_f_* (Ω)	*R_ct_* (Ω)	(*R_f_* + *R_ct_*) (Ω)	*D* (cm^2^ s^−^^1^)
750	3.49 (±0.18)	20.89 (±0.41)	20.40 (±0.81)	41.29	1.41 × 10^−11^
800	5.44 (±0.17)	20.73 (±0.94)	5.69 (±0.10)	26.42	1.59 × 10^−11^
850	5.38 (±0.14)	15.43 (±0.58)	38.45 (±1.12)	53.88	7.99 × 10^−12^
900	9.37 (±0.26)	11.86 (±0.42)	188.60 (±7.73)	200.46	1.27 × 10^−12^

**Table 3 materials-14-01766-t003:** Comparison of electrochemical performance of LiNi_1-x-y_Co_x_Mn_y_O_2_ (0.6 ≤ Ni content < 0.7) materials from different research.

Composition	Preparation Method	Voltage Range (V)	Discharge Capacity (mAh g^−1^)	Capacity Retention	Ref.
LiNi_2/3_Co_1/6_Mn_1/6_O_2_	Rheological phase method	2.8–4.3	188.9 (0.2 C)179.1 (0.5 C)	93.9% (50 cycles, 0.5 C)	This work
LiNi_0.66_Co_0.17_Mn_0.17_O_2_	Sol-gel method	2.5–4.5	169.7 (1 C)	93.8% (25 cycles)	[14]
LiNi_0.65_Co_0.25_Mn_0.1_O_2_	Rheological phase method	2.5–4.5	130.5 (0.125 C)	96.9% (20 cycles)	[16]
LiNi_0.6_Co_0.2_Mn_0.2_O_2_	Solid state method	3.0–4.7	187.9 (0.2 C)	50.6% (150 cycles)	[19]
LiNi_0.6_Co_0.2_Mn_0.2_O_2_	Solid state method	2.7–4.5	179.8 (0.5 C)	69.1% (100 cycles)	[20]
LiNi_0.6_Co_0.2_Mn_0.2_O_2_	Solid state method	2.7–4.5	187.2 (0.2 C)	79.7% (100 cycles)	[21]

## Data Availability

The data presented in this study are available on request from the corresponding author.

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
