# Peer review of "Preparation and Electrochemical Properties of LiNi2/3Co1/6Mn1/6O2 Cathode Material for Lithium-Ion Batteries"

_materials, 2021, doi:10.3390/ma14071766_

Round 1

Reviewer 1 Report

The manuscript entitled „Preparation and Electrochemical Properties of LiNi2/3Co1/6Mn1/6O2  Cathode Material for Lithium-ion Batteries” is considered to be relevant to the scope of this journal.

The authors have made a good synthesis of the literature that provides an overview of the research evolution in this area.

However, several points need to be addressed prior to publication of this manuscript. My comments/suggestions are given:

  1. At the end of the Introduction chapter, the authors must specify very clearly what this paper aims to bring new compared to the results presented in the study of literature.
  2. Why isn't the newly obtained material called LiNi66Co0.17Mn0.17O2 called a notation like those found in the literature? I believe that the same type of material was obtained by other methods by the authors from reference 16.
  3. The number of samples that were analyzed each time must be specified in the experimental chapter.
  4. In order to see the differences between the samples prepared at different temperatures under O2 atmosphere, the SEM images must be presented for the other situations as well. Therefore, I recommend that the authors complete Figure 3 with other SEM images for other samples (obtained at 750, 850 and 900 0C).
  5. Line 177 “The Nyquist plots was performed by fitting the equivalent circuit”. I do not understand. Nyquist diagram is the representation of experimental data. If the fitting with the proposed equivalent circuits was made, it should be very clearly specified which are the experimental data and which are from the fitting.
  6. It is necessary to specify very well the error registered at the fitting for each case. Thus, we can know if the proposed circuit is correct or not.
  7. Also, in figure 4 it is not understood if the Nyquist curves are recorded for samples before or after the charge-discharge tests. Please specify this.
  8. If the authors have an explanation for the fact that the best performance was obtained for the sample calcinated at 800 0C it must be entered in the text. Or at least a correlation with literature data should be made.
  9. Line 215 “Electrochemical performance characterization shows that the sample prepared under the optimized conditions exhibits small interface impedance”. The optimized conditions must be specified very clearly.

Author Response

Comments and Suggestions for Authors

The manuscript entitled „Preparation and Electrochemical Properties of LiNi2/3Co1/6Mn1/6O2 Cathode Material for Lithium-ion Batteries” is considered to be relevant to the scope of this journal.

The authors have made a good synthesis of the literature that provides an overview of the research evolution in this area.

However, several points need to be addressed prior to publication of this manuscript. My comments/suggestions are given:

  1. At the end of the Introduction chapter, the authors must specify very clearly what this paper aims to bring new compared to the results presented in the study of literature.

Response to comment: We thank the reviewer for the careful review and constructive suggestions on our manuscript. The paper aims to provide a new synthesis conditions for preparing high ordered layered LiNi2/3Co1/6Mn1/6O2 material with few defects based on the first-principles computational study [17]. As shown in Table 3, the obtained material shows excellent electrochemical performance compared with some LiNi1-x-yCoxMnyO2 (0.6£Ni content<0.7) materials reported in other papers. We are sorry that that we have not expressed this clearly at the end of the introduction chapter. According to the reviewer’s suggestion, the innovative content of this work is specified at the end of the introduction of the revised manuscript. The added content is as follows:

The present work is to prepare high ordered layered LiNi2/3Co1/6Mn1/6O2 material with few defects by optimizing the calcination temperature and atmosphere based on the first-principles computational study [17], especially the use of oxygen atmosphere. The obtained material shows excellent electrochemical performance compared with some LiNi1-x-yCoxMnyO2 (0.6£Ni content<0.7) materials reported in other papers [16, 18, 21-23]. It can deliver a capacity of 188.9 mAh g-1 at 0.2 C and 130.4 mAh g-1 at 5 C in the voltage range of 2.8-4.3 V, and retain 93.9% of its initial capacity after 50 cycles at 0.5C.

  1. Why isn't the newly obtained material called LiNi0.66Co0.17Mn0.17O2 called a notation like those found in the literature? I believe that the same type of material was obtained by other methods by the authors from reference 16.

Response to comment: We thank the reviewer for pointing this out. We appreciate the opportunity to address the reviewer’s concern. For ternary cathode material LiNixCoyMnzO2, the values of x, y and z can be expressed in fractions or decimals, but x+y+z should be equal to 1. If the fraction becomes a decimal, the sum of the values of x, y and z obtained after rounding cannot be equal to 1, the values of x, y and z are usually expressed in fractions, such as LiNi1/3Co1/3Mn1/3O2. This paper also adopts such an expression, that is LiNi2/3Co1/6Mn1/6O2, which is also prepared according to this composition (As shown in Experimental section). Although the composition of LiNi2/3Co1/6Mn1/6O2 is almost same as that of LiNi0.66Co0.17Mn0.17O2, there are still differences between them. In order to avoid confusion, in the revised manuscript, the composition related to literature 16 are expressed as decimals; and those related to LiNi2/3Co1/6Mn1/6O2 material are all expressed as fractions. Although LiNi0.66Co0.17Mn0.17O2 has been prepared by sol-gel method in literature 16, but its electrochemical performance is unsatisfactory, because its synthesis conditions are difficult to meet the low-defect products. This paper aims to provide a new synthesis conditions for preparing high ordered layered LiNi2/3Co1/6Mn1/6O2 material with few defects based on the first-principles computational study [17]. As shown in Table 3, the obtained material shows excellent electrochemical performance compared with some LiNi1-x-yCoxMnyO2 (0.6 £ Ni content < 0.7) materials reported in other papers.

  1. The number of samples that were analyzed each time must be specified in the experimental chapter.

Response to comment: We appreciate the reviewer’s constructive suggestions and comments. According to the reviewer’s suggestions, the number of samples is specified in the experimental chapter for the characterizations that are not done for all products, e.g. “The microscopic morphologies of the four samples calcinated at different temperatures (750 °C, 800 °C, 850 °C, and 900 °C) were analyzed by Quanta-400F field-emission scanning electron microscopy (FEI, U.S.A)”, “----the electrochemical impedance spectroscopy of the four electrode materials calcinated at different temperatures (750 °C, 800 °C, 850 °C, and 900 °C) after galvanostatic charge-discharge tests----”, etc.

  1. In order to see the differences between the samples prepared at different temperatures under O2 atmosphere, the SEM images must be presented for the other situations as well. Therefore, I recommend that the authors complete Figure 3 with other SEM images for other samples (obtained at 750, 850 and 900 oC).

Response to comment: We appreciate the reviewer’s constructive suggestions and comments. According to the reviewer’s suggestions, the SEM images for other samples (obtained at 750, 850 and 900 oC) were added into Fig.3. In addition, the particle distribution analysis and HRTEM tests were also provided according to the other reviewer’s suggestions. The relative experimental and discussion were revised completely.

  1. Line 177 “The Nyquist plots was performed by fitting the equivalent circuit”. I do not understand. Nyquist diagram is the representation of experimental data. If the fitting with the proposed equivalent circuits was made, it should be very clearly specified which are the experimental data and which are from the fitting.

Response to comment: We thank the reviewer for the thoughtful and critical comments, and we have followed advice to address the points raised by the reviewer. We have remade Figure 5. In the Figure 5 (b), the dots (■, ●, ▲, ▼) represent the experimental data, and the lines are drawn with the data obtained by fitting the equivalent circuit. The sentence “The Nyquist plots was performed by fitting the equivalent circuit” has also been revised to be “The Nyquist plots was fitted by using the equivalent circuit (Fig.5b inset),------”.

  1. It is necessary to specify very well the error registered at the fitting for each case. Thus, we can know if the proposed circuit is correct or not.

Response to comment: We sincerely thank the reviewer for pointing this out, and we have followed the advice to specify the error registered at the fitting for each case. The error of each fitting value is shown as follows:

T (℃)

Error(%)Re

Error(%)Rf

Error(%)Rct

750

5.1196

1.9782

3.9562

800

3.0982

4.5184

1.7349

850

2.5442

3.7471

2.9136

900

2.7279

3.5814

4.0996

It can be seen that the error of each fitting value is less than 6%, far less than 10% (this value is a general requirement for impedance fitting error), which shows that the proposed equivalent circuit is correct. To simplify the expression, the error is converted into the impedance difference and shown in Table 2 (in parentheses) in the revised manuscript.

  1. Also, in figure 4 it is not understood if the Nyquist curves are recorded for samples before or after the charge-discharge tests. Please specify this.

Response to comment: We thank the reviewer for pointing this out. We appreciate the opportunity to address the reviewer’s concern. Actually, we have clearly stated in the experimental chapter that the electrochemical impedance spectroscopy of the four electrode materials were tested after galvanostatic charge-discharge tests at 0.2C for 5 cycles.

  1. If the authors have an explanation for the fact that the best performance was obtained for the sample calcinated at 800 oC it must be entered in the text. Or at least a correlation with literature data should be made.

Response to comment: We sincerely thank the reviewer for the constructive suggestions. In this work, the study on the influence of the calcination temperature showed that the samples calcined at 800 oC under an oxygen atmosphere exhibits the best electrochemical performance. In order to further prove its superior electrochemical performance, comparison with LiNi1-x-yCoxMnyO2 (0.6£Ni content<0.7) materials reported in other papers was also done in this paper. As shown in Table 3, the material still shows excellent electrochemical performance. In order to express more clearly that the best performance was obtained for the sample calcinated at 800 oC, a sentence was revised in the paragraph for discussion of Table 3. It is that “Compared with LiNi0.66Co0.17Mn0.17O2 [16] and LiNi0.65Co0.25Mn0.1O2 [18] materials obtained using a similar wet chemical synthesis route, the LiNi2/3Co1/6Mn1/6O2 sample calcinated at 800 ℃ under O2 atmosphere in this work exhibits excellent electrochemical performance.”

  1. Line 215 “Electrochemical performance characterization shows that the sample prepared under the optimized conditions exhibits small interface impedance”. The optimized conditions must be specified very clearly.

Response to comment: We appreciate the reviewer’s constructive suggestions and comments. According to the reviewer’s suggestions, a sentence in the conclusion chapter has been revised. It is “ XRD and SEM analysis results indicate that the sample produced under the optimized calcination conditions of 800 ℃ and oxygen atmosphere displays well-crystallized particle morphology, high ordered layered structure with low defects.”

Reviewer 2 Report

Authors produced nickel-rich ternary layered LiNi2/3Co1/6Mn1/6O2 materials by rheological phase method. They studied the effect of calcination temperature and ambient on the crystal structure of the obtained material. Although the number of characterization is low, the manuscript is well written and clear, however I have some comments and questions for the authors:

  1. Row 135: how did they calculate the I003/I104 ratio? from area or height? Probably they refined the lattice parameters by MDI-JADE software, as described later in the manuscript, in this case I suggest to move the sentence closer to this row.
  2. row 144: I suggest to write also discharge capacity and columbic efficiency value of the material produced in air, in order to underline the differences between the two materials.
  3. Authors presented SEM image for only the material produced at 800°C, they do not know the morphologies of the others materials, therefore they cannot say that this morphology help to provide higher capacity. I suggest to show SEM images for materials obtained at 750, 850, and 900°C too.

Author Response

Authors produced nickel-rich ternary layered LiNi2/3Co1/6Mn1/6O2 materials by rheological phase method. They studied the effect of calcination temperature and ambient on the crystal structure of the obtained material. Although the number of characterization is low, the manuscript is well written and clear, however I have some comments and questions for the authors:

  1. Row 135: how did they calculate the I003/I104 ratio? from area or height? Probably they refined the lattice parameters by MDI-JADE software, as described later in the manuscript, in this case I suggest to move the sentence closer to this row.

Response to comment: We thank the reviewer for the careful review and comments on our manuscript. We appreciate the opportunity to address the reviewer’s concern. In this work, the intensity ratio of (003) and (104) diffraction peaks, referred as I(003)/I(104), is calculated from the height according to the usual method in the study of ternary LiNi1-x-yCoxMnyO2 cathode materials. The height of diffraction peaks can be obtained from the peak search report using MDI-JADE 6.5 software package. According to the reviewer’s suggestions, a sentence “The obtained XRD data can be analyzed using the MDI-JADE 6.5 software package” has been added to a position closer to Row 135.

  1. row 144: I suggest to write also discharge capacity and columbic efficiency value of the material produced in air, in order to underline the differences between the two materials.

Response to comment: We appreciate the reviewer’s constructive suggestions and comments. According to the reviewer’s suggestions, the sentence in row 144 was revised to be “Compared with the sample produced in air, its discharge capacity at 0.2C rate in the range of 2.8-4.3 V is increased from 170.8 mAh g-1 to 188.7 mAh g-1, and the coulombic efficiency is also increased from 82.1% to 88.3%.”

  1. Authors presented SEM image for only the material produced at 800°C, they do not know the morphologies of the others materials, therefore they cannot say that this morphology help to provide higher capacity. I suggest to show SEM images for materials obtained at 750, 850, and 900°C too.

Response to comment: We appreciate the reviewer’s constructive suggestions and comments. According to the reviewer’s suggestions, the SEM images for other samples (obtained at 750, 850 and 900 oC) were added into Fig.3. In addition, the particle distribution analysis and HRTEM tests were also provided according to the other reviewer’s suggestions. The relative experimental and discussion was revised completely.

Reviewer 3 Report

  1. The authors recommend using the correct stoichiometric notation for LiNi2/3Co1/6Mn1/6O2.
  2. Please correct Fig. 1a and Fig. 2 Y-axis units and provide the plot's standard JCPDS data.
  3. The author should discuss the importance and consequence of the c/a ratio in the XRD section.
  4. The authors are strongly recommended to provide the TEM and HR-TEM data to claim “the clear crystal planes and grain boundaries” as mentioned above Fig. 2, lines 157-158.
  5. Please provide the particle size distribution diagram.
  6. Why cross-over potential is high in Fig. 1b and Fig. 4a?
  7. Why references appeared in the conclusion section?
  8. English must be improved throughout the manuscript. There are many awkward and grammatically incorrect expressions.

Author Response

Comments and Suggestions for Authors

  1. The authors recommend using the correct stoichiometric notation for LiNi2/3Co1/6Mn1/6O2.

Response to comment: We thank the reviewer for the careful review and comments on our manuscript. We appreciate the opportunity to address the reviewer’s concern. For ternary cathode material LiNixCoyMnzO2, the values of x, y and z can be expressed in fractions or decimals, but x+y+z should be equal to 1. If the fraction becomes a decimal, the sum of the values of x, y and z obtained after rounding cannot be equal to 1, the values of x, y and z are usually expressed in fractions, such as LiNi1/3Co1/3Mn1/3O2. This paper aims to provide a new synthesis conditions for preparing high ordered layered LiNi2/3Co1/6Mn1/6O2 material with few defects based on the first-principles computational study [17]. We adopt the stoichiometric notation for LiNi2/3Co1/6Mn1/6O2 due to the fact that the sum of the values of x (0.67), y (0.17) and z (0.17) obtained after rounding cannot be equal to 1. In addition, LiNi2/3Co1/6Mn1/6O2 was also prepared according to this composition (As shown in Experimental section) in this work.

  1. Please correct Fig. 1a and Fig. 2 Y-axis units and provide the plot's standard JCPDS data.

Response to comment: We thank the reviewer for pointing this out. According to the reviewer’s suggestions, we have remade Fig. 1a and Fig. 2, in which Y-axis unit has been corrected, and the plot’s standard JCPDS data has been provided.

  1. The author should discuss the importance and consequence of the c/a ratio in the XRD section.

Response to comment: We appreciate the reviewer’s constructive suggestions and comments. According to the reviewer’s suggestions, the importance and consequence of the c/a ratio are discussed in the revised manuscript, which is “It is well known that in addition to the intensity ratio of I(003)/I(104), the c/a value is also used generally to indicate the cation mixing, and a higher ordered layered structure is obtained when c/a > 4.899 [36]. That means that----” . A relevant reference [36] has also been added.

  1. The authors are strongly recommended to provide the TEM and HR-TEM data to claim “the clear crystal planes and grain boundaries” as mentioned above Fig. 2, lines 157-158.

Response to comment: We appreciate the reviewer’s constructive suggestions and comments. According to the reviewer’s suggestions, TEM and HR-TEM data (Fig. 4) have been provided in the revised manuscript. The relative experimental and discussion have also been revised completely.

  1. Please provide the particle size distribution diagram.

Response to comment: We appreciate the reviewer’s constructive suggestions and comments. According to the reviewer’s suggestions, the particle size distribution diagram (Fig. 3e) has been provided in the revised manuscript. The relative experimental and discussion has also been revised completely.

  1. Why cross-over potential is high in Fig. 1b and Fig. 4a?

Response to comment: We thank the reviewer for the careful review and comments on our manuscript. We appreciate the opportunity to address the reviewer’s concern. It is well known that there are usually two ways to express the charge-discharge curve of lithium-ion battery materials. One is that the charge curve and the discharge curve are continuous in time, and a graph without cross-over points will be obtained, e.g. Figure 12 (A) and (B) in the reference (Journal of The Electrochemical Society, 2017, 164 (14), A3529-A3537 ).

Another is that both the charge curve and discharge curve start from zero respectively, and thus the charge and discharge curve will cross. This method is most commonly used. In this work, this method is used to draw the charge-discharge curves, and the corresponding cross-over potentials are about 3.75V. The values are very close to the results of Figure 3 (a) in the reference (Journal of Power Sources, 2013, 233, 121-130).

  1. Why references appeared in the conclusion section?

Response to comment: We thank the reviewer for pointing this out. Based on the first-principles computational studies [16, 17], LiNi2/3Co1/6Mn1/6O2 is considered to be very promising cathode materials. This work confirmed experimentally that LiNi2/3Co1/6Mn1/6O2 material with few defects can be prepared by optimizing the synthesis conditions based on the first-principles computational study [17], and can exhibit excellent electrochemical performance. Because the first-principles computational analysis came from the references, so we wrote the references in the conclusions. In the revised manuscript, the references were deleted, and the last sentence of conclusions was revised to be “The results fully prove---”.

  1. English must be improved throughout the manuscript. There are many awkward and grammatically incorrect expressions.

Response to comment: We sincerely thank the reviewer for pointing this out, and we are very sorry for our English writing. However, we are always doing our best to try to improve. Based on your helpful and valuable comments, we have revised the entire manuscript to our best.

Round 2

Reviewer 1 Report

The authors have made all the required corrections so that the manuscript can be published in its current form.